# 3D Modeling of a Virtual Built Environment Using Digital Tools: Kilburun Fortress Case Study

**Ihor Tytarenko [1], Ivan Pavlenko [2,\*] and Iryna Dreval [1]**

[1] Department of Urban Planning, O.M. Beketov National University of Urban Economy in Kharkiv, 17 Marshal Bazhanov St., 61002 Kharkiv, Ukraine

[2] Department of Computational Mechanics Named after Volodymyr Martsynkovskyy, Sumy State University, 2 Rymskogo-Korsakova St., 40007 Sumy, Ukraine

\* Correspondence: i.pavlenko@cm.sumdu.edu.edu.ua

**Abstract:** The reliable reconstruction of cultural or historical heritage objects is an urgent problem for humanity. It can be successfully solved using up-to-date 3D modeling tools. The proposed technique allows for modeling virtual environments at an even higher level. This study aimed to develop an approach for designing historical heritage objects with sufficient accuracy using a built environment. The Kilburun Fortress was chosen as the object of study. The modeling procedure includes monitoring the object's territory, analyzing archival, librarian, and cartographic sources, and further modeling and reproducing the research object in a virtual environment using various software tools. The following stages were implemented during this study: analysis and processing of preliminary data (analysis of plans and schemes, overlapping maps); the scaling of graphical objects for the reliable reproduction of the studied object; the design of a working 3D model using AutoCAD and SketchUp; the rendering and final processing of textures using Quixel; and visualization using Twinmotion. As a result, a model of the historical heritage object was created using 3D means. The model can also be integrated into ArchiCAD and Revit software.

**Keywords:** 3D modeling; reconstruction; infrastructural development; architectural building; rendering; sustainable land management

## 1. Introduction

Today, the global scientific society continues to develop up-to-date building information modeling (BIM) technologies supplemented by corresponding software for designing buildings and structures [1]. There are many topical areas where these technologies are widely used. This especially applies to the reconstruction of destroyed objects of cultural and historical heritage [2].

Lack of investments, a careless attitude, and permanent natural destruction are the main challenges of long-term reconstruction. However, BIM technology allows for overcoming these challenges due to the use of a virtual environment. The related approaches allow for implementing research works more rapidly, speeding up their efficiency and quality.

Recently, a number of state-of-the-art studies in modeling virtual environments have highlighted the significance of the considered problem. Particularly, Ramos Sánchez et al. [3] proposed an approach for achieving universal accessibility by the remote virtualization and digitization of complex archaeological features. As a result, a methodology for heritage management under a collaborative digital environment was realized for the case study of Columbarios in the UNESCO World Heritage City of Mérida, Spain.

Meoni et al. [4] applied information modeling techniques to integrate structural performance monitoring in historical buildings. The developed software integrates monitoring data using historical building information modeling (HBIM) for buildings located in the countryside of Perugia, Italy. Rodrigues et al. [5] applied a deep learning approach to classify buildings' degradation. As a result, a workflow for automating recognition defects

in historical buildings was developed for the Igreja da Misericórdia de Aveiro case study in Aveiro, Portugal.

Aricò and Lo Brutto [6] applied scan-to-BIM and HBIM to model an ancient Arab-Norman church in Palermo, Italy. The proposed method allows for converting a parametric model into a virtual environment. As a result, the geometrical representation of building elements and the integration of different data types were enhanced. Banfi et al. [7] applied laser scanning and photogrammetric means to the semantic modeling of artifacts. As a result, research progress in identifying quality methodologies for representing, interpreting, and modeling complex contexts was summarized for the Claudius-Anio Novus Aqueduct case study in Tor Fiscale, Roma, Italy.

Sarıcaoğlu and Saygı [8] proposed a data-driven conservation technique for heritage places curated with HBIM. As a result, digital data-driven conservation actions were introduced by implementing a novel methodology for ancient building remains at the Erythrae archaeological site, Izmir Province, Turkey. Liu et al. [9] applied ArcGIS 10.8.2 software to create a 3D model of spatial pedigree for the Jiangnan region in China based on the on-site review of the village space.

Lee et al. [10] applied terrestrial laser scanning for the 3D data acquisition of indoor assets. As a result of the photogrammetry application, it was shown that even if the modeling process was not fully automated, the results of the 3D modeling were accurate.

Liu et al. [11] developed a semantic and context information fusion network for view-based 3D model classification and retrieval. As a result of using ModelNet10, ModelNet40, and ShapeNetCore55, a compact 3D representation was obtained. Moreover, the superiority of the proposed method compared to the state-of-the-art on both 3D classification and retrieval tasks was demonstrated.

Ciski et al. [12] applied a geographical information system (GIS) to develop a sustainable heritage management system. As a result of various map content generalizations, the importance of data generalization in spatial modeling was found, and the acceptable level for the data's loss of accuracy was generalized.

Luchetti et al. [13] applied texture mapping to create high-quality photorealistic 3D virtual reality (VR) integration. Joel et al. [14] discussed the advantages of the joint use of BIM, GIS, and VR integration to manage infrastructural objects. Additionally, Barrile [15] developed an innovative tool for 3D model reconstruction of the cultural heritage. The proposed experimental HBIM processing approach allowed for the reconstruction of the "Conventazzo" in San Pietro di Deca, Torrenova, Messina, Italy.

Fernández-Mora et al. [16] analyzed different approaches for integrating structural projects into the BIM paradigm. In addition, Farzaneh et al. [17] analyzed various design processes for building energy modeling (BEM). As a result of considering recent technological developments in this field, the main research gaps in the BIM-BEM process were partially eliminated.

Azzam et al. [18] proposed an integrated approach for sustainability assessment in power plant projects using BIM. As a result, the analytical hierarchy process was implemented to evaluate the weight factors considered in the proposed sustainability system. Karmakar et al. [19] implemented an automated route planning approach for construction sites utilizing BIM. As a result, the efficiency of daily equipment movements at the site was increased by reducing possible conflicts and enhancing accessibility.

Li and Wang [20] developed a green building integrated evaluation system based on the BIM environment. The results proved the evaluation efficiency of such an approach for green building evaluation. Zhong and Qin [21] applied laser image scanning technology in the environment. As a result, a complete architecture and detailed process application were established for China's construction industry.

Li and Wang [22] studied green construction technologies and their practical application for designing large bridges in mountain areas. Additionally, Wang et al. [23] demonstrated a construction planning technique for bridges in mountainous areas based on virtual construction technology. As a result, this virtual construction technology was

used for 3D planning and the intelligent control of access roads for the case study of southwestern China. Potseluyko et al. [24] developed an interactive environment using VR for the self-build housing sector.

Ishizawa [25] applied the machine learning-based clustering algorithm to the project environment. Shaharuddin [26] analyzed digital twin approaches for smart city purposes to provide numerous benefits for enhanced situation assessment, decision-making, coordination, and resource allocation. As a result, methods for 3D modeling and the virtual replication of buildings were systematized. Abdelrahman et al. [27] proposed models using digital twins for preference prediction with spatial–temporal proximity data from Build2Vec software. Evangelou et al. [28] also built digital twins of smart cities for a two-story building located in the city of Larisa in northern Greece.

A few novel approaches have been widely used for designing new buildings. Particularly, Yin et al. [29] proposed a technique for semantic localization on maps using a 3D LiDAR sensor. As a result, the feasibility of the proposed mapping-free localization framework was proven with reduced localization errors. Muñoz et al. [30] developed an approach to modeling a sports pavilion in Seville, Spain. Three-dimensional modeling was carried out using Edificius and CYPECAD MEP software to design the building's facilities.

Vermandere et al. [31] applied a two-step alignment of mixed-reality Hololens devices to project planning, construction monitoring, and maintenance. Liu and Wang [32] studied the green performance of university buildings in northern China by combining the aspects of planning, building design, system design, energy management, and energy conservation planning.

Do Carmo et al. [33] developed a framework for architecture and structural engineering collaboration in BIM projects through structural optimization. The obtained results allowed for solving technological barriers related to software interoperability. Schischmanow et al. [34] realized seamless navigation, 3D reconstruction, and thermographic and semantic mapping for building inspection for the case study of a house with surroundings in the village of Morschenich near the city of Jülich, North Rhine-Westphalia, Germany.

Ragnoli et al. [35] implemented a multi-technological architecture for construction site monitoring using the long-range wide-area network (LoRaWAN) and radio frequency identification (RFID) techniques. As a result, a comprehensive system of structural nodes. along with an RFID access management system and LoRaWAN gateway features, was provided. D'Amico et al. [36] implemented an integrated approach for healthy buildings. As a result, an architectural design based on indoor air quality prediction was developed. Finally, Zhao et al. [37] applied a multi-objective optimization technique for low-carbon and energy-saving buildings. As a result, an energy and optimization design and the developed genetic algorithm allowed for conserving power and reducing emissions for existing buildings.

Finally, Pocobelli et al. [2] thoroughly analyzed the main methodologies for built heritage science. They mentioned that today's BIM does not allow for fully automated procedures to model heritage buildings, which also emphasizes the importance of improving existing approaches in 3D modeling for designing historical heritage objects.

Based on the abovementioned studies, the following research objectives were formulated. Firstly, analysis and processing of preliminary data were carried out. Secondly, graphical objects were appropriately scaled. Next, a working 3D model was designed. Finally, textures were processed and rendered.

All these objectives allowed for achieving the main aim of the research—to develop an approach for recreating objects of historical heritage with sufficient accuracy using up-to-date built environments.

## 2. Materials and Methods

During this study, methods of observation, analysis, and comparison and a reproduction of the research object in a 3D built environment using appropriate software were

applied. The corresponding design scheme for implementing the research methodology is shown in Figure 1.

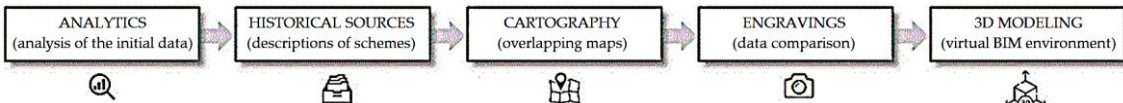

**Figure 1.** Research stages.

In this study, the built environment considered architectural objects that harmoniously fit into the existing environment's surroundings and adapted them to reproduce the historical heritage.

The main research stages included analysis of the initial data, detailed descriptions of plans and schemes, overlapping ancient maps onto modern ones, data comparison and analysis, and creating a flexible model in a virtual environment.

The research on the available sources was as follows: analytical studies of maps were carried out (cartographic sources of the 16–17th centuries, modern satellite images in Google Maps, studying the terrain using topography and visual analysis), and descriptions (archival descriptions of eyewitnesses to the events, engravings, drawings, photos, detailed explanations for object plans) and materials (actual search for the remains of building materials, analysis of the building principles, obtaining information about materials) were collected. The results of the analytical experiments were modeled by creating a flexible 3D model, which allowed for the reproduction of an object of cultural and historical heritage. This model enabled changing properties, which can be conducted with any model (e.g., using Revit software), and visualization was achieved using Twinmotion.

The design scheme of the proposed approach is presented in Figure 2.

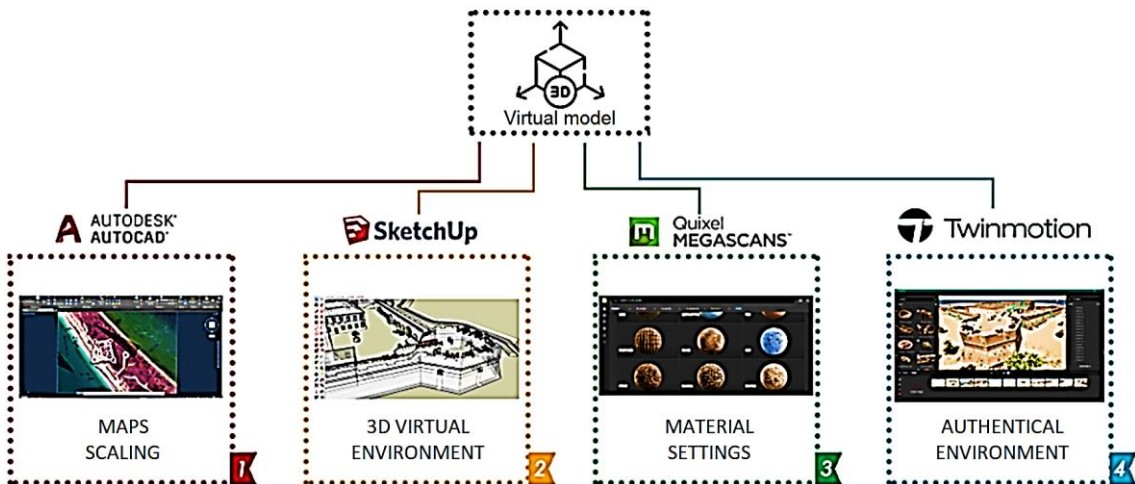

**Figure 2.** The design scheme of the applied approach.

As a case study, the architectural renovation of the Kilburun Fortress at Kinburn Spit in Ukraine was considered (Figure 3). Kinburn is the site of an ancient settlement of people of the Bronze Age. Nowadays, this is a sandy peninsula with a length of 40 km and an average width of about 8–10 km, which the Black Sea washes on one side. The Kinburn spit separates the Dnipro Bug Estuary and Yagorlytsky Bay.

The fortress was an essential strategic object since it controlled the entrance of the trade routes of the Dnipro and the Southern Bug. However, the fortress was destroyed after massive battles in 1787 and during the Crimean war. Nowadays, it is one of the most mysterious and understudied buildings.

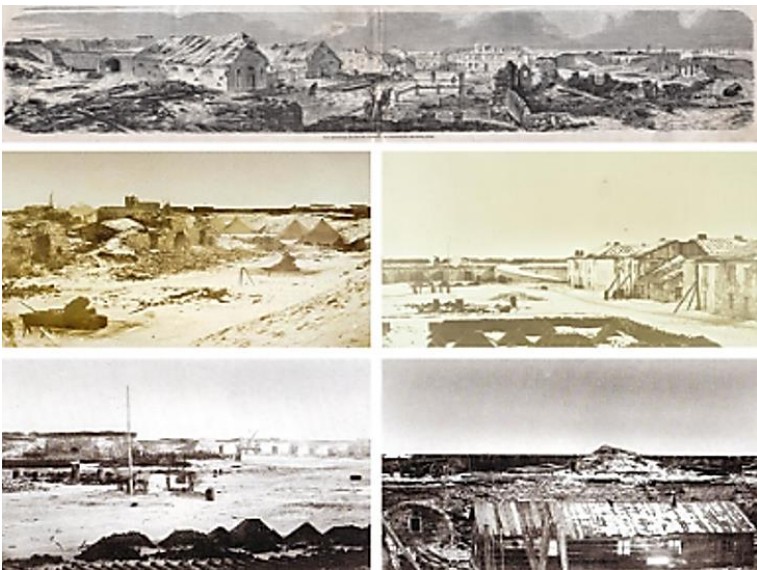

**Figure 3.** The Kilburun Fortress.

Remarkably, the territory of the Kinburn Spit is unique not only in its status as a reserve but also in its historical significance for Ukraine, Turkey, and Greece.

During modeling, the following obstacles were considered. Firstly, maps and diagrams were scaled appropriately for the reliable reproduction of the 3D object. For this purpose, Autodesk software tools (e.g., AutoCAD) were applied.

Secondly, SketchUp was also used for the 3D model creation, considering the application of many different formats. Quixel software can also be used due to its open-access library of materials, their properties, and photo scans. Such software allows for obtaining a photorealistic rendering. After creating the model using the appropriate tools, it was integrated into Revit and ArchiCAD software.

Finally, the machining time, as a primary operability factor, was considered for effective modeling. Corresponding software (e.g., Twinmotion, Corona Renderer, V-Ray) can accelerate the rendering process.

The method was also supplemented by applying a topogeodetic survey, e.g., using photogrammetry technology. This technology makes the process faster and more informative than the classical method using ground geodetic measuring devices. The stereotopographic method for creating an accurate topogeodetic subbase was applied for this purpose. Firstly, photos of the studied area according to the appropriate scheme were taken and aligned using Agisoft Metashape software. In this case, the model's accuracy was up to 3 cm in width and height. Secondly, the process of creating a cloud of points was started and completed. Thirdly, a digital relief model was created (e.g., according to the methodology described in [38]), and an orthophoto plan was realized using Digitals software. Afterward, the conversion of the images for further processing of the contours based on aerial phototriangulation was realized. Finally, the created shapes were combined with the final plan using AutoCAD software.

## 3. Results

### 3.1. Analysis and Processing of Preliminary Data

Before processing the results, documentation and historical and cartographic sources were analyzed. Before starting the modeling, a historical study of the object and searches for sources (e.g., archive and librarian) were carried out. After such an analysis, the corresponding schemes were created, including overlaying the maps and determining the location. Since the available ancient schemes of the 16–17th centuries do not contain correct coordinate systems, the analysis was mainly based on finding maps and scheme matches.

Next, a field survey and site study was carried out (e.g., relief features, remains fixation). At this stage, it is also advisable (if available) to use photogrammetry and laser scanning of the site, e.g., using Agisoft, 3DF Zephyr, Meshroom, or RealityCapture.

After the analysis of archival, librarian, and cartographic sources, identification of the researched object location was carried out. In this case, the image processing stages were as follows. Firstly, the studied maps were loaded into the program environment to overlay them further. Next, the necessary transparency of the images was adjusted. After applying the free transformation tools, the scheme of the studied object was superimposed on the actual terrain, considering the remains of fortifications and defensive lines. In the presence of a dimensional ruler, there was an additional opportunity to adjust the scaling accuracy. The subsequent stage concerned cleaning up the object plan layer using the tool for removing the white background.

The image processing results are presented in Figure 4.

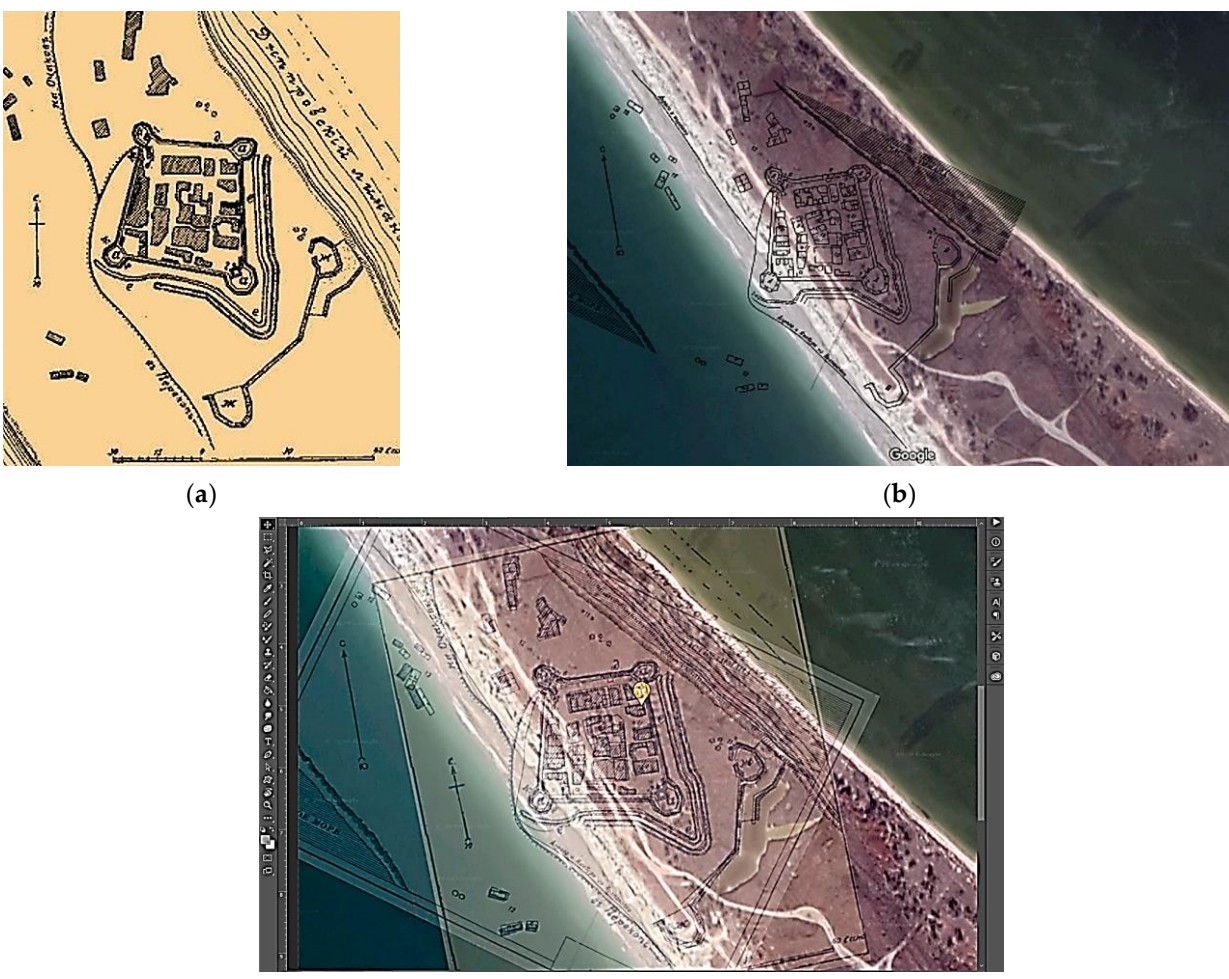

**Figure 4.** Analysis (**a**), overlay (**b**), and processing (**c**) of maps.

### 3.2. Scaling of Graphical Objects

Scaling up the research object is also an important stage. It allows for the development of a more accurate 3D model by reaching the required dimensions of the building for further research. However, the accuracy of this method depends on the quality of the descriptions and schemes available in the archival and cartographic sources. For this purpose, computer-aided design (CAD) supporting image formats and scaling functions are applicable.

However, photo scanning from drones can also be used to provide geodetic measurements. In this case, a model was created based on the scanning points. Particularly, a

geodesic base was created using Agisoft Metashape. In other methods, geodetic measurements with a theodolite and leveling can be applied.

However, superimposing ancient maps and schemes on the created ones can also be applied. In this case, the topological and geodetic data were based on the scale using corresponding software (e.g., ArchiCAD, Revit).

The sequence of this stage was as follows. Firstly, the edited map in *.jpg format was imported into AutoCAD software. Using the scale ruler of Google Maps, the exact distance from conventional points A and B was identified. After using geometric tools, a line segment along with the known distance was measured. The same line segment tool, another line, but already with the natural scale of 1:1, was drawn. After applying the alignment tool, the image and the first line was scaled relative to the second line due to the known value in the natural scale. Finally, the file in *.dwg format was saved for further processing.

The sequence of carrying out this research stage is presented in Figure 5.

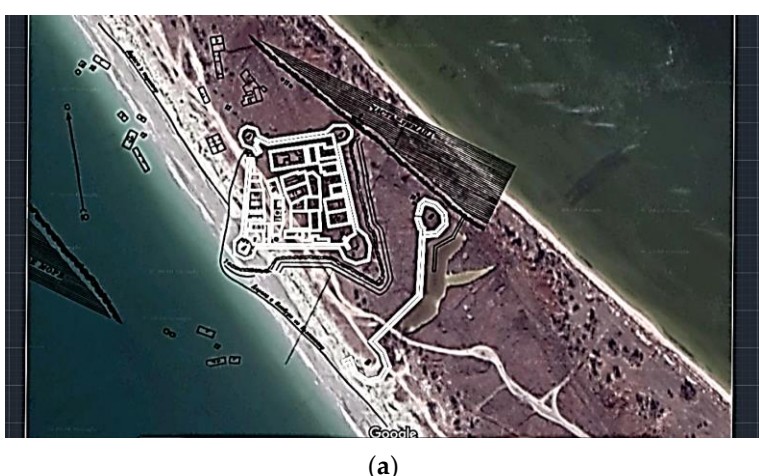

(**a**)

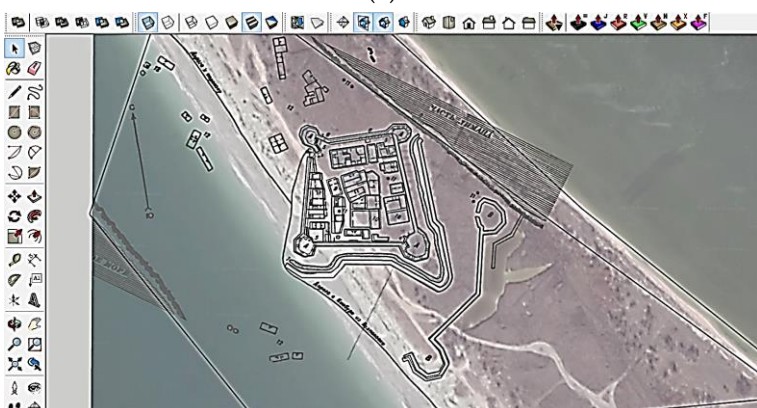

**Figure 5.** Scaling (**a**) and processing (**b**) of the summary map.

### 3.3. Creating a Working 3D Model

The next stage was to create a working 3D model using SketchUp software tools. Due to the known scale evaluated in the previous stage, it was possible to create a high-quality model for further processing.

The sequence of this stage was as follows. The *.dwg file was imported into SketchUp software. Next, the object was modeled. Mainly, historical elements were added using open-source 3D libraries. Due to the related functions, it was possible to obtain the technical data of the studied object (Figure 6).

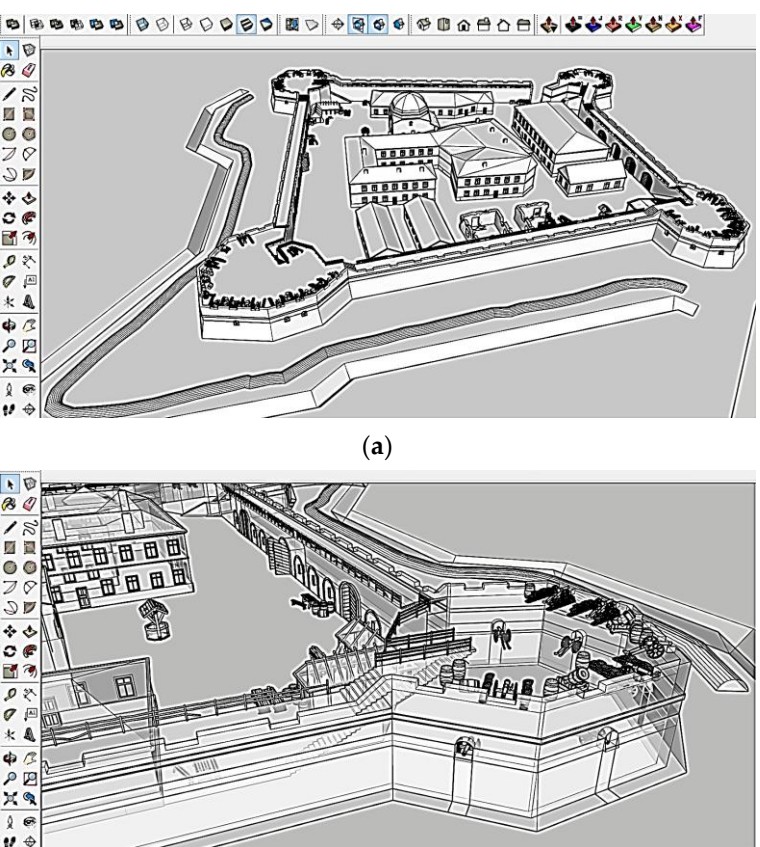

**Figure 6.** The general 3D model of the object (**a**) and its parts (**b**).

To create a terrain based on the data of geodetic measurements, Bézier curves and splines were applied as SketchUp tools. For designing contours, walls, holes, and other features, the tools for creating and editing geometric objects, styles, and projection modes were used. Such an approach for object modeling can also be applied in combination with Revit and ArchiCAD.

### 3.4. Processing of Textures and Rendering

Upon completion of the 3D modeling, the previous model was imported into Twinmotion software based on Unreal Engine for the final processing of textures and fast high-quality rendering. Additionally, due to the available Quixel library, it was possible to select authentic materials and their properties for the final virtual environment. Additional 3D elements of the environment are also added, and the light and weather conditions are adjusted. Finally, rendering scenes were configured.

The results of the material selection for creating an authentic environment and further rendering are presented in Figure 7.

### 3.5. Final Modeling and Visualization

As a result of realizing the Kilburun Fortress, a flexible 3D model was created, allowing for further modification and improvement to reproduce the historical heritage building in real life.

The applied method of object modeling can also be used in combination with programs such as Revit and ArchiCAD. The data imported into Revit can also be used instead of SketchUp. This allows for various properties of the model.

The architecture visualization was realized using Twinmotion software and its library of textures and models to achieve photorealistic rendering. Due to plugins, it can also be combined with SketchUp, Revit, and ArchiCAD.

The overall results are presented in Figure 8.

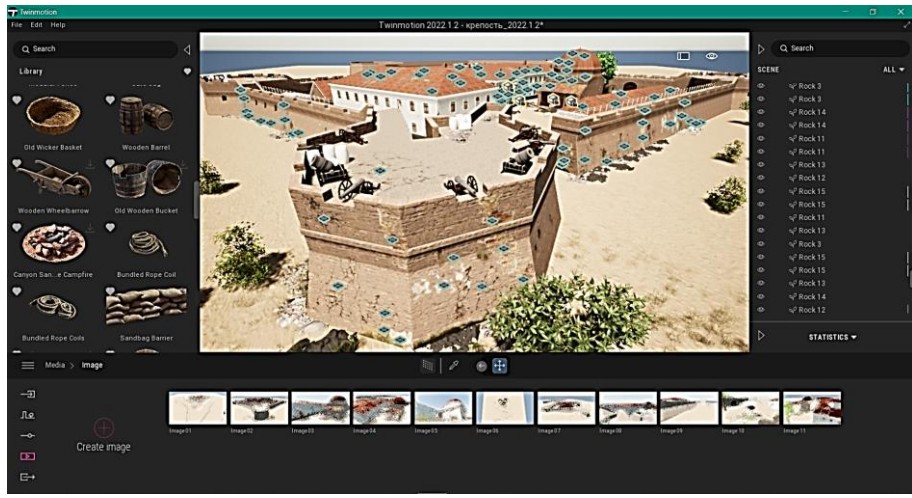

(**a**)

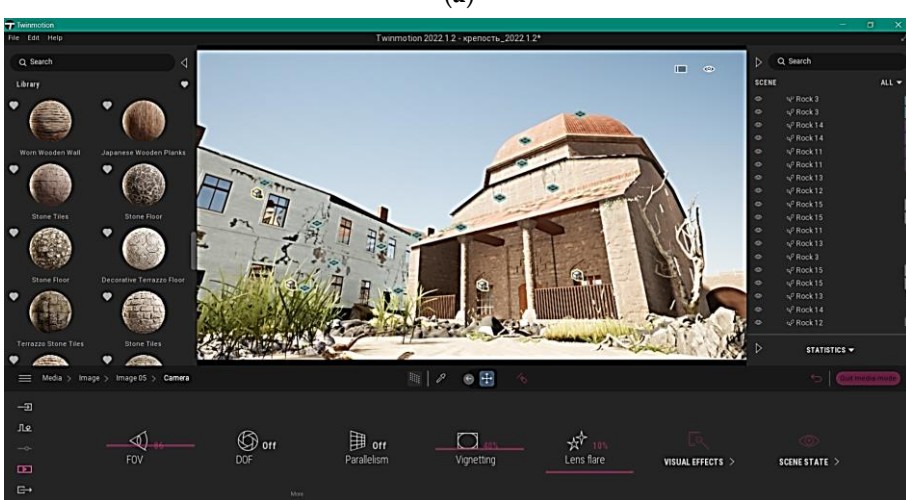

**Figure 7.** Creation of an authentic environment (**a**) and rendering (**b**).

The Kilburun Fortress was considered the research object in this study. Nevertheless, the application of this method is not limited to studies of fortification structures. Since the object has a historical background, the designed structure is influenced by the archaeological and cartographic research results. Today, the historical reconstruction of fortresses aims to preserve the historical heritage. Simultaneously, fortifications for military affairs lost their relevance after the appearance of rifled artillery.

The approach is promising in studying and creating a model of studied objects and can be applied and integrated with BIM technologies. With the application of this method, other historical objects have also been created (e.g., the reconstruction of the Sumy Fortress, a guard tower in Mogrytsia of Sumy Region, and the Yeni-Dünya Fortress near Hacibey in Odesa Region).

This method is planned to soon be applied in studying medieval fortresses in the Northern Black Sea region in the South of Ukraine. The development of such projects will primarily be aimed at preserving historical, archaeological, and architectural buildings and reconstructing Ukrainian cities in the post-war period.

The practical significance of the approach can also be highlighted by the architectural heritage strategy in Ukraine, mainly according to the national restoration program initiated by the government. This program aims to restore a large number of historic architectural buildings.

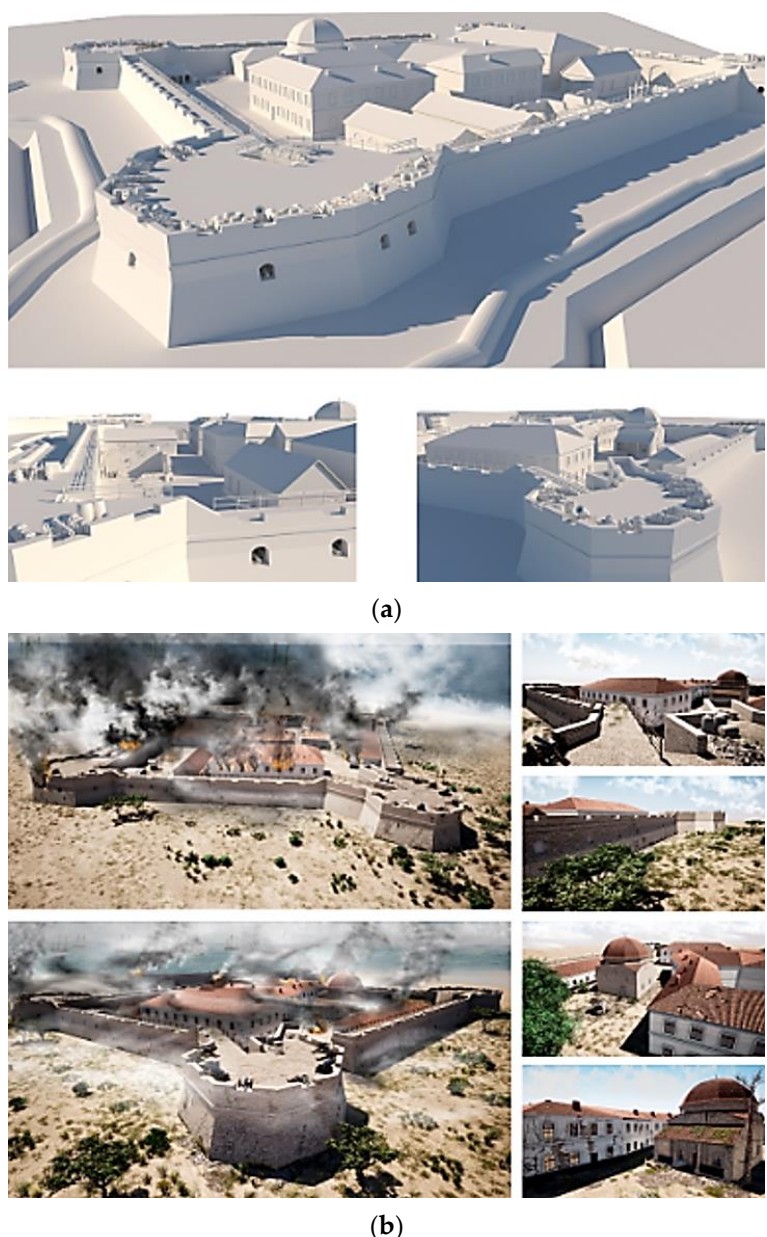

(**a**)

(**b**)

**Figure 8.** The final model (**a**) and its visualization (**b**).

## 4. Discussion

Using multiple software tools and analyzing the related programs' applications provided an opportunity to combine relevant approaches to reproducing a research object, with the 3D reconstruction of the Kilburun Fortress as a case study. As a result, fortification structures and buildings of the 18th century were recreated, considering the analysis of authentic terrain based on the study of ancient maps.

Based on the developed approach, the created model provides an opportunity to reproduce the studied object as closely as possible. Additionally, it has complete information on the configuration of the object, dimensions, volumes, and materials, depending on the quality of the historical and archaeological sources and the authenticity and reproduction accuracy. Thus, the complete information about the research object makes it possible to reproduce it in real life.

The applied research methods have the following advantages compared to other studies: fast use of the software tools, a relatively short processing time, the creation of a model using other sources to complement the modeled object (i.e., ancient maps, plans,

diagrams, and drawings, photo sources), availability of the complete information in the 3D model of the studied object, and integration with other programs.

Since there are no limits to virtual environment applications for building objects [39], the proposed approach can supplement GIS, which still has disadvantages in terms of its expensiveness, complexity in integration with maps, and higher storage capacity, but has relatively high accuracy and allows for exploring extensive data in a wide variety of visualization forms [40].

Another advantage of the proposed model is the use of affordable, easy-to-learn programs with a wide application range of various tools for creating a 3D model. After using the applied programs, it is possible to analyze schemes, create terrains based on geodetic measurements, and create an accurate model for further reconstruction and visualization.

According to the up-to-date experience of other studies [11,12], it is possible to combine the proposed approach with methods based on photogrammetry, photo scanning, and laser scanning [10]. This allows for the development of a more detailed model of the built environment and the creation of geodetic measurements in the relevant programs.

The accuracy of the proposed method was reached by a detailed study of the studied area, comparison, and the overlaying of historical maps on modern topographical and satellite ones. It also depended on the scaling of the maps and the detailed descriptions of archival and historical sources, plans, and schemes. All this made it possible to reproduce the location of the investigated object with acceptable accuracy. However, the incompleteness of the archaeological data increased the research's complexity.

A design model of the object of historical heritage was created using up-to-date 3D software. Initially, ancient and modern maps, archival descriptions, engravings, and drawings were analyzed, and possible methods of their integration into Revit or ArchiCAD software were indicated for each stage of the modeling.

Overall, the applied approach can be used as an algorithm to solve the topical issue of preserving architectural buildings [15,41] and reconstructing historical heritage objects [2]. It also allows for the use of the model in information sources, textbooks, historical overviews, and visualizing the researched built environment with further implementation into VR. Additionally, using the current programs and methods, it is possible to design a virtual model for further reconstruction and develop technical solutions proven for centuries in these buildings and implement them for a new experience.

## 5. Conclusions

As a result of this study, a design model of the object of historical heritage was created using up-to-date 3D software. Initially, ancient and modern maps, archival descriptions, engravings, and drawings were analyzed. As a result, additional aspects were evaluated to improve the model (e.g., observation and consulting, 3D modeling, and search for supplemental materials and objects to fill the built environment of the historical heritage object).

During the research, the actual problem of reliable reconstruction of the Kilburun Fortress as a European historical heritage was solved by applying modern software tools for 3D modeling, including monitoring of the object state, data analysis, and further reproduction of the object.

As a result, a virtual model of the historical heritage was created by processing the initial data, scaling the graphic objects, processing texture, and performing a final rendering. The main design stages included map scaling by Autodesk, building a 3D virtual environment using SketchUp, material selection using Quixel Megascans, and creating the final authentical environment using Twinmotion software.

Overall, after applying 3D modeling, a virtual environment for the Kilburun Fortress case study was created, and an integrated approach to designing historical heritage objects based on architectural environments has been developed.

Further research will be aimed at implementing modern software tools for creating VR, the application of which will allow for improving architectural buildings and highlighting the problems of preservation and restoration of ancient heritage sites. In the

future, it will be possible to model and reconstruct ancient and modern objects based on historical authenticity.

Moreover, the post-war reconstruction of Ukraine is highly essential, not only for residential infrastructure but also for destroyed historical objects. In this regard, reconstructing historical heritage objects (castles, estates, parks, historical museums, etc.) is also essential. For this purpose, recommendations to further develop exchange information requirements (EIR) to optimize the management of information content among the various stakeholders and on the methods of verification, storage, and delivery of models according to the standards ISO 19650 and PAS 1192-2 will be proposed further.

**Author Contributions:** Conceptualization, I.T.; methodology, I.T.; software, I.T. and I.D.; validation, I.P.; formal analysis, I.P. and I.D.; investigation, I.T.; resources, I.T.; data curation, I.D.; writing—original draft preparation, I.P. and I.T.; writing—review and editing, I.D.; visualization, I.T.; supervision, I.D.; project administration, I.P.; funding acquisition, I.P. All authors have read and agreed to the published version of the manuscript.

**Funding:** This research received no external funding.

**Data Availability Statement:** The data presented in this study are available on request from the corresponding author.

**Conflicts of Interest:** The authors declare no conflict of interest.

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
