# Peer review of "3D Modeling of a Virtual Built Environment Using Digital Tools: Kilburun Fortress Case Study"

_applsci, doi:10.3390/app13031577_

Round 1

Reviewer 1 Report

The paper exposes the development of a 3d model with graphic treatment to reconstruct the Kilburun fortress, which constitutes a relevant effort and a good example of historical representation, but its contribution is weak for a scientific journal of international scope. The process is well described, with appropriate references and proper use of software tools, but the title is confusing due to the mention of BIM, which is a multidisciplinary construction management and coordination methodology, but in this case it is not it deals with execution, only shape and visual expression, nor do different disciplines participate. Therefore, the use of software only to model geometry is not the same as applying a BIM methodology. Nor is a critical analysis of the process or its implications expressed, which findings are noted for other similar applications. The conclusions only welcome the model created, like a development report, but with little deductive depth. Likewise, there is no review or discussion of its architectural or construction characteristics that could contribute to historical or technical knowledge. It is suggested to analyze this experience and the recreated building with an expressive and historical sense.

Author Response

We appreciate all your valuable comments and suggestions, which allow us to strengthen the manuscript and improve its representation. The article has been revised significantly. Also, please find attached all our answers.

Reviewer 2 Report

-     - The article is well written, however it seems to have a very narrow perspective. The word “comprehensive” is misleading. The title should rather use the more realistic term “case study”. The result is not at all “comprehensive”, since almost no alphanumeric data or BIM standards (IFC, bsDD, BCF, ISO19650, LOIN,…) are used. The authors present a very classic 3D Model in VR, not BIM

- The authors should make clear, why they use the term BIM. A sharper line should be drawn to purely graphical models (3D,VR,..) and proper semantic information models.

- The authors have to explain the difference between the term “Architectural Environment” and “Built Environment” as commonly used in ISO19650

- How have the "Exchange Requirements" been shared between the stakeholders? How will the "BIM" model be used in future?

- The Introduction contains a bunch of randomly chosen publications.  The authors should only consider publications that are relevant for their own research. The authors do not contribute any scientific progress to AI, AR, IoT or route planning. Why should the cited literature be relevant/related to this paper? Also there should be a rigid order, for example: Research in data acquisition with laser scanning/photogrammetry, semantic information models and classification, GIS integration in heritage management, Texture Mapping, VR/AR integration, Specific Tools for HBIM…

- Line 150: The authors must explain what they mean by a flexible/adjustable 3D model?

- Section 2 and Section 3: The used material has to be quantified in detail: The authors have to add number/structure/quality of used documentation, coordinate reference system, surveying equipment with precision and accuracy, numerical documentation of the scaling, including residuals of the test data, validation procedures for textures, used geometric modeling tools from SketchUp,… and so on.

- The sections 4. Discussion and 5. Conclusion have almost the same content. There is no critical discussion on workload, tools, data formats, …. Please discuss also the limitations of your technology stack (CAD/GIS/VR). What would you do ifferently in the future?

Please forgive my harsh judgement. As a reviewer, however, I have the task of checking scientific standards. This includes correct terminology, as well as quantifiable details and critical discussions of your own results.

Author Response

We appreciate all your valuable comments and suggestions, which allow us to strengthen the manuscript and improve its representation. The article has been revised. Also, please find attached all our answers.

Reviewer 3 Report

Authors are advised to follow suggestions and revised the paper accordingly as per attached file.

 Manuscript Title:

A Comprehensive Approach to Modeling the Architectural Environment Based on BIM

The topic of the manuscript is one of the most appreciable domain of research now-a-days and particularly the region selected needs these type of researches in the current scenario.

The manuscript titled: “A Comprehensive Approach to Modeling the Architectural Environment Based on BIM” is informative and presented a good research in the domain of digital applications to Heritage Objects. The manuscript presented good approach for digital applications to the irreplaceable resource of our past. The abstract reflected the detailed process followed by the authors for carrying out the said research and overall conception of the research. Unfortunately, the authors said objectives were not fully achieved with the formulated methodology. The literature review in general presented more on BIM applications in various domains rather than focused literature required for the said topic. The manuscript also needs English corrections and proof reading to be considered for further processing. The detailed review is as follows:

Detailed Review:

Title: The title should reflect the content in the manuscript and the present title emphasized the BIM tool which was not utilized for the stated objectives. Keeping the same content of the manuscript the proposed titled should be like this just an example: A Comprehensive Approach to Modeling the Architectural Environment through Digital Tools Application 

Abstract: The abstract is concise and clear. However, the last sentences are highlighted in the manuscript that needs English corrections. In addition to that the conclusion of the manuscript should be mentioned as a closing sentence.

Introduction: This section is well written and mapped the literature relative to BIM applications to Heritage sector. The BIM tools application was clearly presented and explored with various additional features as set in objectives of the research.

However the Literature Review section merged with the introduction was not found in continuation with the introductory part. Rather authors were found distracted in outlining the benefits of BIM in various other domains. The literature map needs to be focused on the BIM usage in Heritage domain. 

Materials and Methods: The section is well written but authors were not able to state the BIM modelling software’s placement and usage in the methodology or in the whole process. I would like to recommend this article to authors to know the BIM modelling for heritage science.

“BIM for heritage science: a review”

Authors defined BIM model is actually the application of digital tools for heritage reconstruction modelling. Basic software is Autodesk Revit used for BIM modelling.

Results: The section is well written and explained. The whole process for digital applications is followed and stepwise concluded as well. Again the BIM approach was not found in this section.

“Overall, a design model of the object of historical heritage has been created using up-to-date 3D software and BIM technologies. Initially, ancient and modern maps, archival descriptions, engravings, and drawings were analyzed. As a result, additional aspects were evaluated to improve the model”

This quoted section also not clearly presents the BIM technologies used in the whole process.

Conclusions: The comments are the same as no BIM approach was found and instead of this, if authors just focus on the digital applications wording would reflect the manuscript content.

In the end authors are advised to restructure the manuscript keeping in view the proposed title of application of digital tools in replacement of BIM technologies to justify the content of the manuscript.

Author Response

We appreciate all your valuable comments and suggestions, which allow us to strengthen the manuscript and improve its representation. The article has been revised. Also, please find attached our answers.

Round 2

Reviewer 1 Report

It is valued that the authors have corrected the paper adapting the sense of the work carried out, completing a literature review and extending the conclusions, but it is suggested to deepen the analysis and criticism for a substantial contribution to the academic community. In particular on the use of digital representation tools, their eventual advantages of speed and exchange for data recording and future developments. The possibility of revising the geometric composition of the fortifications is an issue that should be discussed, which combines the topographical situation, the capacity of the weapons and the space occupation. As well as regarding the materiality and its current deterioration, if the representation can reproduce the original conditions or different recovery alternatives. As a support strategy for architectural heritage decision-making.

Author Response

Thank you for your evaluation. According to your suggestion, it should be noted that the Kilburn Fortress has been considered as a research object within the approach. Nevertheless, its application is not limited to studies of fortification structures. Since the object has a historical background, the designed structure is influenced by the archaeological and cartographic research results. Today, the historical reconstruction of fortresses aims to preserve the historical heritage. Simultaneously, fortifications in military affairs lost their relevance after the appearance of rifled artillery. Moreover, the approach is promising in studying and creating a model of the studied objects, which can be applied and integrated with BIM technologies. With this method’s application, other historical objects have also been created (e.g., reconstruction of the Sumy for-tress, guard tower in Mogrytsia of Sumy Region, and Yeni-Dünya fortress near Hacibey in Odesa Region). Soon, this method is planned to be applied in studying medieval fortresses in the Northern Black Sea region in the South of Ukraine. The development of such projects will primarily be aimed at preserving historical, archaeological, and architectural buildings and reconstructing Ukrainian cities in the post-war period. The practical significance of the approach can also be highlighted by the architectural heritage strategy in Ukraine, mainly according to the national restoration program initiated by the government. This program aims to restore a large number of historic architectural buildings. The corresponding explanations have been added at the end of the results.

Reviewer 2 Report

I'm still missing a proper geodetic analysis of accuracy, georeferencing, specific research questions ...but OK

Author Response

Thank you for reviewing the article and understanding. According to your suggestion, it can be noted that the method can also be supplemented by applying a topogeodetic survey, e.g., using photogrammetry technology. This technology makes the process faster and more informative than the classical method using ground geodetic measuring devices. However, the stereotopographic method for creating an accurate topogeodetic subbase should be applied for this purpose. Firstly, photos of the studied area according to the appropriate scheme are taken and aligned using the Agisoft Metashape software. In this case, the model’s accuracy is up to 3 cm in plan and height. Secondly, the process of creating a cloud of points is started, followed by completing the process. Thirdly, a digital relief model is created (e.g., according to the methodology described in [39]), and an orthophoto plan is realized using the Digitals software. After, the conversion of images for further processing of contours based on aerial phototriangulation is realized. Finally, the created shapes are combined with the final plan using the AutoCAD software. The corresponding explanations have been added at the end of the materials and methods.

Reviewer 3 Report

Authors revised the manuscript but needs some English corrections and proof reading

Author Response

Thank you for the review. The English language has been corrected more thoroughly.
